# Bacterial Infections Associated with Immunosuppressive Agents Commonly Used in Patients with Interstitial Lung Diseases

**DOI:** 10.3390/pathogens12030464

**Published:** 2023-03-15

**Authors:** Said Chaaban, Ruxana T. Sadikot

**Affiliations:** 1Pulmonary, Critical Care and Sleep Medicine, Department of Internal Medicine, University of Nebraska Medical Center, Omaha, NE 68198-6450, USA; 2VA Nebraska-Western Iowa Health Care System, Omaha, NE 68105, USA

**Keywords:** interstitial lung disease, immunosuppressants, pulmonary fibrosis, sarcoidosis, infection, review

## Abstract

There are about 200 different types of interstitial lung disease (ILD), and a crucial initial step in the assessment of a patient with suspected ILD is achieving an appropriate diagnosis. Some ILDs respond to immunosuppressive agents, while immunosuppression can be detrimental in others, hence treatment is based on the most confident diagnosis with consideration of a patient’s risk factors. Immunosuppressive medications have the potential to result in substantial, and perhaps life-threatening, bacterial infections to a patient. However, data on the risk of bacterial infections from immunosuppressive treatment specifically in patients with interstitial lung disease is lacking. We hereby review the immunosuppressive treatments used in ILD patients excluding sarcoidosis, highlight their risk of bacterial infections, and discuss the potential mechanisms that contribute to the increased risk of infections.

## 1. Introduction

The term “interstitial lung diseases” (ILDs) refers to a group of lung diseases that may cause diffuse remodeling, architectural damage to healthy lung tissue, and progressive loss of lung function [1]. ILD is essentially a parenchymal lung disease with restriction that presents as diffuse infiltrative shadows. The two most frequent ILDs seen in clinical practice are idiopathic pulmonary fibrosis (IPF) and sarcoidosis; however, there are well over 100 other types of ILD that have been discovered based on clinical presentation, radiographic features, and histopathologic investigation [1]. Some types of ILD may remain stable or even spontaneously improve, while others may deteriorate over time, but may respond to immunosuppression [2]. Clinicians must determine whether to treat a patient with an immunosuppressive agent after reaching a confident diagnosis and evaluating patients’ clinical comorbidities [2]. While immunosuppression can be the mainstay of treatment for some forms of ILDs, one of the major concerns is the altered host immune response that predisposes to infections.

Immune suppressive therapies range from widely used drugs such as corticosteroids, to newer agents, such as Rituximab, that demonstrate effects by targeting specific molecules on immune cells. Corticosteroids quickly replaced other treatment modalities for a range of diffuse lung illnesses since their first use back in 1948. At that time, it was believed that corticosteroid use in ILD could stop the progression to irreversible fibrosis, respiratory failure, and ultimately death, since steroids are known to have anti-inflammatory properties that reduce the formation of granulation tissue [1]. The NIH-sponsored IPF Network study (PANTHER) shows that immunosuppressive therapies for the chronic treatment of idiopathic pulmonary fibrosis (IPF), one of the idiopathic interstitial pneumonias, does not appear to have a significant impact on this disease, and may even be harmful [3]. The findings of this trial show significant mortality and a hospitalization risk for patients with IPF who were in the azathioprine plus N-acetylcysteine (NAC) treatment arm compared to the placebo and NAC-only arms [2,3]. One of the major concerns is a risk of increased infection with high dose steroids used in these patients [3]. Here, we will review the potential mechanisms by which steroids can increase the risk of infection and the clinical studies that have tried to define the relationship between infection in those who are treated with steroids.

With recent advances and a multidisciplinary approach to diagnose and treat ILD, there has been an increase in the use of biologics and newer immunosuppressive agents [2]. In general, the mechanisms by which these agents exhibit immune suppressive effects include myelosuppression and macrophage and lymphocyte inhibition, which becomes a double-edged sword. Because these are key immune cells, the suppression of pathways to modulate the disease process may result in the inhibition of their protective effects to combat infections. Rituximab, one of the biologic agents, targets CD20 single cell surface molecule found on B lymphocytes. Several new agents that target specific receptors or surface markers are in development. Therapeutic agents that target immunosuppressive cells, either by direct inhibition or altering their functions, also decrease the barrier to effective immune response, rendering the host susceptible to infections. Because the use of these agents has the potential to result in substantial, and perhaps life-threatening, consequences that can cause serious injury to patients, the decision to use them is both challenging and risky. Therefore, it is essential for treating professionals to have a thorough understanding of both the side effects of immunosuppressive medications and the monitoring techniques recommended by clinical practice standards. Careful monitoring can help avoid negative outcomes [2]. There are compelling data in the medical literature to support the use of immunosuppressive medications to treat connective tissue disease-associated (CTD)-ILD (e.g., CS, mycophenolate, other disease modifiable factors), and other forms of ILD [2]. However, there are very few adequately powered, prospective, randomized, placebo-controlled, double-blind clinical trials to evaluate the safety and efficacy of these medications and to compare them head to head [2]. Numerous immunosuppressive medications used to treat immune responsive ILDs have been thoroughly tested for safety and effectiveness in clinical trials for the treatment of different types of CTD vasculitis, inflammatory bowel disease, or post-transplant rejection of solid organs [2]. This review will focus on the immunosuppressive medications that have been used in different forms of ILDs, excluding sarcoidosis, and the risk of bacterial infection posed by their use. It is important to emphasize that most of the literature used in this review is extrapolated from other diseases that utilize the same therapies that have been described in ILD. There is very scarce data on bacterial infection risk in ILD patients who are on immunosuppression therapy. There also may be a confounding risk of the underlying disease affecting the risk for bacterial infection that cannot be elucidated without more trials.

### Methods

We performed a literature review on bacterial infections, immunosuppression, and interstitial lung disease, accessing articles published since 1976. We chose the 58 that are most relevant and with no redundancy of information for inclusion in this review.

## 2. What Immunosuppressive Therapies Are Used for ILDs?

In several different ILDs, corticosteroids are an effective treatment. Acute eosinophilic pneumonia [4,5], chronic eosinophilic pneumonia [6,7], cryptogenic organizing pneumonia [8,9], cellular nonspecific interstitial pneumonia (NSIP) [10], and hypersensitivity pneumonitis [11] generally respond effectively and often rapidly to corticosteroids. Drug-induced pneumonitis also improves with a combination of corticosteroids and stopping the offending agent, especially if the bronchoalveolar lavage (BAL) fluid mostly results in lymphocytes or eosinophils [1]. Corticosteroids are typically used in concert with other immunosuppressive drugs in the treatment of ILD secondary to connective tissue disease, even though controlled data are sparse [1]. The BAL of diseases that are most responsive to corticosteroids are eosinophilic or lymphocytic. In general, conditions that tend to have a neutrophilic predominance respond poorly to steroids [12].

### 2.1. Steroids

Steroids have multiple anti-inflammatory and immunosuppressive effects, and can affect any immune cell [13]. The mechanisms by which steroids exhibit immune suppressive effects include the inhibition of macrophage differentiation and the synthesis of proinflammatory prostaglandins and leukotrienes, interleukin 1, interleukin 6, and tumor necrosis factor [13]. They also inhibit the tumoricidal and microbicidal functions of activated macrophages [13]. Additionally, steroids prevent neutrophils from adhering to endothelial cells and hinder the release of lysosomal enzymes, respiratory burst, and chemotaxis to the area of inflammation [13]. All lymphocyte subpopulations are susceptible to marked lymphopenia secondary to glucocorticoids, which also prevent T-cell activation by blocking interleukins 2, 3, and 4 [13]. This also affects the maturation of T lymphocytes and immunosuppresses the formation and operation of dendritic cells [13]. Corticosteroids can increase the risk of infection in a dose-dependent manner as studies suggest that there is a dose-dependent effect on the immune system [14]. T lymphocyte counts are decreased at dosages of 2 mg/kg (CD4+ > CD8+), and at higher doses (>2 mg/kg) there is inhibition of lymphocyte activation and the B cell production of antibodies [14].

The typical dosage of prednisone for immunosuppression in the ILD patient is on average 0.5 to 1 mg/kg ideal body weight per day for four weeks, followed by 30 to 40 mg/day for 8 weeks. If patients respond and/or stabilize, then the dosage is slowly tapered, with the aim to reach a minimum (such as 5 mg) every other day, with a goal to discontinue use after a year. Treatment may be for a longer period, with a range of 17.4 ± 12.1 months [15,16]. The precise duration or even whether there is a need for indefinite treatment has not been described. Additionally, there is a lack of consensus with regards to the optimal timing to add a cytotoxic agent. The addition of a steroid sparing agent has been described at diagnosis, with progression or if the patient is deemed dependent on corticosteroid therapy [15].

The risk of infection increases with dosage and treatment duration, but tends to stay low in patients exposed to low doses, even with high cumulative dosages [14].

Most of the available information about the risk of infections associated with corticosteroids comes from randomized controlled trials and observational studies (both population-based and single/multicenter) [13]. Unfortunately, most of these studies are limited by the small sample size, making it difficult to interpret studies and harder to extrapolate results to a larger population. Additionally, the underlying disease itself may contribute to a decline in immunity, and patients may be on other immunosuppressive drugs; thus, identifying a direct causal relationship between studies and the risk of infections with steroids is difficult [17].

A metanalysis of 71 clinical trials revealed a relative risk (RR) of 1.6 in infections in patients treated with corticosteroids compared to those who were not on corticosteroids. The RR increased to 2.0 when the dosage was between 20 and 40 mg per day. There was no difference if patients were on <10 mg a day or with a cumulative dose of <700 mg of prednisone [17]. The highest risk for serious infection, as shown by the metanalysis by Dixon et al., was with patients who received 30 mg/day of prednisone for one month (adjusted odds ratio (OR) 1.84; 95% confidence interval (CI) 1.58–4.00) compared to those who were exposed to 5 mg/day for six months (adjusted OR 1.46; 95% CI 1.31–4.65) [18]. Corticosteroid therapy can increase the risk of any pyogenic bacteria, such as Staphylococcus bacterial infections, and because steroids tend to mask the effects of an infection, patients tend to present at a later stage of their infection [17,19]. Conn and Poynard’s metanalysis noted that the sepsis rate was 6.5% (2868 patients) in the corticosteroids arm compared to 4.8% (2776 patients) in the placebo arm [20]. Nazareth et al., in their population-based cohort study, demonstrate a relative increased risk for lower respiratory tract infection during the first weeks of steroids use, and that risk decreased past this point [21].

From a phenotypic standpoint, the increased risk of infection was associated with age, diabetes, dosage of steroids, and hypoalbuminemia [21]. Furthermore, patients with asthma, COPD, and cancer carried a higher risk of septicemia [21]. Durand and Thomas note in their study the following adjusted rate ratios for lower respiratory tract infections (1.48, 95% CI 1.34–1.65), urinary tract infections (1.27; 95% CI 1.34–1.65), and sepsis (1.63; 95% CI 0.78–3.40) in 1664 patients who received corticosteroid therapy for Giant cell arteritis compared to 8078 control patients [22]. Post-surgical pneumonia and sepsis was more pronounced in patients with systemic lupus erythematosus (SLE) who were exposed to corticosteroids [23]. There was also a fourfold increase of serious bacterial infections in patients with inflammatory bowel disease who were on corticosteroid therapy [24]. Bernatsky et al. show in their case control study of more than 23,000 patients that nearly 16% (906/5529) of patients who received steroids for rheumatoid arthritis (RA) suffered from pneumonia [25].

The risk of tuberculosis (TB) reactivation increases with prolonged therapy using moderate- or high-dose corticosteroids [21]. Based on the dosage found to suppress the tuberculin skin test in earlier studies [21,26,27,28], the Centers for Disease Control and Prevention (CDC) and the American Thoracic Society determined that a 15 mg daily dose of prednisone for one month is the threshold for increased risk. The adjusted hazard ratio for tuberculosis in one case-control study was 2.8 (95% confidence range, 1.0–7.9) for those taking less than 15 mg of prednisone per day versus 7.7 (95% confidence interval, 2.8–21.4) for those taking more than 15 mg per day (or equivalent) [29]. Hence, patients receiving moderate- to high-dose corticosteroids are regarded as being at higher risk of latent TB becoming active. The lowest dose of prednisone that increases this risk is unknown [21,26,27,28].

In addition to the direct effects of glucocorticoids on immune response, the suppression of adaptive immunity may result in response to vaccination. Firstly, individuals using glucocorticoids may not mount a vaccination response, leaving them defenseless from the diseases that vaccines are meant to prevent. Secondly, people who receive live vaccinations may experience active illnesses. The current guidelines seek to lower the likelihood of both potential problems [21,30,31,32,33]. Due to the limits of the available data, all published recommendations for immunization of immunocompromised hosts are based on expert opinion. Additionally, existing recommendations vary significantly across countries [21].

The guidelines given by the American Society of Transplantation, the Advisory Committee on Immunization Practices, the European League Against Rheumatism, the Infectious Disease Society of America, and the American Society for Blood and Marrow Transplant are noted next [30,31,32,33,34,35]. Patients who require moderate to high doses of corticosteroids (20 mg/day of prednisone) for at least 2 weeks should be questioned about their vaccination history to ensure that they are current on the following immunizations: Haemophilus influenza B, Neisseria meningitides, Streptococcus, and tetanus toxoid [19,36]. The recommendations are that if at all feasible, patients who are immunized prior to starting corticosteroid therapy should undergo immune surveillance to ensure that the proper protective immune response takes place. Inactivated vaccinations may be safely administered to patients who must begin therapy with moderate to high doses of systemic glucocorticoids right away, with the caveat that their eventual immunity may be subpar. However, live vaccines should not be administered to anyone taking corticosteroid doses equal to 20 mg of prednisone per day for more than 2 weeks. The administration of live vaccines (such as those for oral typhoid, bacillus Calmette-Guerin, and yellow fever) in this clinical scenario should be postponed until the patient has been taking prednisone doses of 20 mg per day (or equivalent) for at least 1 to 3 months [19,31,37].

The pneumococcal pneumonia vaccine should also be given to all people aged less than 65 and over who are immunosuppressed (including those taking dosages of prednisone >20 mg/day). The pneumococcal vaccine should also be encouraged for patients taking chronic low-dose steroids, especially if they are also taking steroid-sparing medications. According to the recommendations of the Advisory Committee on Immunization Practices, the 13-valent pneumococcal vaccination (PCV13), followed by a dose of the 23-valent vaccine (PPSV23), should be given to patients who have never had a pneumococcal vaccine. A second dose of PPSV23 is advised after five years. Patients who have had PPSV23 vaccinations in the past, but not PCV13, should have the PCV13 one year after [30,33].

Together, these studies highlight the increased risk of infections in patients who receive steroids, including their inability to respond to vaccines. Patients who need steroids for prolonged periods should be screened and monitored for infections and receive the appropriate immunization to prevent infections.

### 2.2. Steroid Sparing Agents and Combination Therapy

Treatment of many of the interstitial lung diseases is becoming challenging and in many cases combination therapy is needed to treat these patients [38]. A recent large case series of patients with CTD-ILD demonstrates a tendency toward disease stabilization and possible improved lung function as well as the ability to significantly reduce corticosteroid dosing with combination therapy [38]. Cytotoxic agents are typically combined with corticosteroids with the aim of controlling the disease and reducing the corticosteroid dose to help avoid side effects [2]. Which agent and combinations of agents are used, the duration of treatment, and the timing of selected agents is based on expert opinion confined to case series and uncontrolled clinical trials rather than guidelines and recommendations [39].

The management of the patient with CTD-ILD usually targets the immune system, which is associated with the production of autoantibodies associated with CTD. Azathioprine was examined in three retrospective trials in patients with CTD-ILD and was shown to have similar efficacy compared to mycophenolate with the stability of lung function testing [39,40].

Multiple randomized controlled trials in systemic sclerosis associated with ILD note that cyclophosphamide compared to placebo is tolerable, and mycophenolate is non-inferior [40,41]. In the SCENSIS trial, patients who were on mycophenolate alone or in combination with nintenadib witnessed a smaller decline in lung function testing compared to those who were on mycophenolate alone [42]. Tociluzimab and Rituximab have shown promising results in the treatment of systemic sclerosis with ILD [40]. The use of combination therapy is a harbinger of advanced interstitial disease, which further increases the risk of infection. We hereby review the immunosuppressants that have been used in ILD, excluding sarcoidosis, and point out their risk of bacterial infections in the following sections.

### 2.3. Azathioprine

Azathioprine (AZA) leads to myelosuppression and to the inhibition of B and T cell proliferation. Patients with leukopenia secondary to AZA are susceptible to bacterial infections [14,43]. In a retrospective trial, Boerner et al. show that 20% (11/56) of patients suffered from an infection, however only 5% (3/56) had to discontinue treatment secondary to infection [40]. In a larger case control study, which included 23,000 patients with rheumatoid arthritis, an increased risk of infection with the use of AZA compared to MTX is described. Patients who were taking AZA show a moderate to increased risk of severe infections, with 20% (52/259) of patients exposed suffered from pneumonia [25]. Another study described that AZA is associated with 12% (3/25) risk of infection with otitis media, necrotizing pneumonia and cellulitis described [43]. These data indicate that the risk of bacterial infection is significant in patients who are treated with AZA.

### 2.4. Mycophenolate Mofetil

Mycophenolate, an inosine monophosphase dehydrogenase inhibitor that acts by decreasing T and B cell proliferation via a decrease in purine synthesis, leads to leukopenia [14,44]. In a study by Dheda et al. in which they treated patients with systemic sclerosis and ILD, Mycophenolate was associated with a nearly 5% risk of infection. However, it should be noted that a high percentage (94.4%) of patients where this risk was described were also concomitantly treated with prednisone (mean dose 8.22 mg/day) [44]. Kingdon et al. retrospectively studied the safety of mycophenolate in biopsy-proven lupus on 13 patients. Most patients were on concomitant steroids. In their cohort, 23% (3/23) of patients had infections. Infections described were salmonella gastroenteritis, staphylococcal abscess that required drainage, and respiratory failure [45]. Mycophenolate use in patients with scleroderma ILD is also associated with the risk of respiratory infection [41]. Overall, the data on mycophenolate and infections are scant, as most of the studies include patients who were on a combination of steroids and mycophenolate, making it difficult to ascertain the effects of the individual agents. Further studies are needed to define mycophenolate and infection risk.

### 2.5. Cyclophosphamide

High-dose cyclophosphamide has recently been used in the treatment of ILD, especially those resulting from an autoimmune mechanism [46]. The use of cyclophosphamide can cause myelosuppression (leukopenia, neutropenia, thrombocytopenia, and anemia), bone marrow failure, and severe immunosuppression, which may lead to serious and, sometimes, fatal infections. Neutropenia and lymphopenia associated with the use of cyclophosphamide can lead to an increase in the susceptibility to infection, mainly gram-positive and gram-negative infections [14]. Few studies provide details of infection occurrence in patients receiving cyclophosphamide. Sehgal et al. report that significant numbers of patients on cyclophosphamide developed neutropenia (96%), of whom 68% developed clinical infectious complications [46]. The majority of the bacterial infections described include streptococcus, staphylococcus, enterococcus, and C difficile. The risk of infection with the use of cyclophosphamide is related to neutropenia in addition to the underlying disease [47].

### 2.6. Rituximab

Rituximab is a chimeric human monoclonal antibody that is directed against the CD20 antigen on B lymphocytes [48]. As a result of binding to CD20, B lymphocytes, with the exception of plasma cells, are depleted through complement and/or antibody-dependent cellular cytotoxicity [14]. B-cell depletion lasts for 6 to 9 months or longer, and is associated with possible hypogammaglobulinemia [14]. Not only that, but after the depletion of the B cells, the new B cells that are produced are immature rather than memory B cells. The delayed development of memory B cells lasts for years following the last injection of Rituximab [49]. If the hypogammaglobulinemia is severe, then the patient is at risk for bacterial sinusitis and pneumonia [14]. Patients who had higher levels of IgG levels at baseline had less risk for infections. Thus, it has been proposed that providers consider IV Ig at levels of <5 g/L prior to Rituximab infusion, especially in patients who have a history of serious infection [49].

The risk of severe infections increases within two months from the first dose. Patients who were at increased risk and had serious infections were older, had concomitant diabetes mellitus, lower CD 19 counts, lower immunoglobulin levels, renal failure, and were continued on corticosteroids (>5 mg/day). Infections are mostly bacterial and the most common bacterial pathogens were pseudomonas Aeruginosa, Escherichia coli, and staphylococcus aureus [49]. Patients who are vaccinated against S. pneumoniae have a lower risk of infection [49]. Pneumococcal vaccinations 3–4 weeks before the first dose of Rituximab is preferred and having all live attenuated vaccines updated is also recommended [49].

### 2.7. Abatacept and Tociluzumab

Abatacept, approved for the treatment of moderate to severe RA, is a fusion protein made of cytotoxic T-lymphocyte-associated protein 4 and a crystallizable fragment portion of IgG1 that leads to the inhibition of T-lymphocyte co-stimulation. It is mainly used in patients that have not had a positive response to TNF inhibitors and disease-modifying antirheumatic drugs (DMARDs). There is growing evidence regarding its utility in RA-ILD. Compared to other therapies, it carries a good safety profile and is associated with a low risk for serious infections [50]. The risk of hospitalized infections and TB were not different when compared to patients receiving DMARDS [51]. More studies investigating infections associated with abatacept are needed to be able to elucidate its overall risk for bacterial infection [50].

Tocilizumab is an anti-IL 6 monoclonal antibody that has been used in the treatment of CTD ILD and has been approved by the Food and Drug Administration for the treatment of systemic sclerosis-associated ILD [52]. Infections documented with use of Tocilizumab include pneumonia, infective tenosynovitis, sepsis, and otitis media [53]. Future studies will inform about the risks related to infections with the use of Tocilizumab.

### 2.8. Prophylaxis

A mitigation strategy to help decrease the risk of some bacterial infections while patients are on immunosuppressive therapy is vaccination [54,55]. We reviewed vaccination use in corticosteroids earlier. Table 1 summarizes vaccination use in patients on immunosuppressive therapy is presented below [54,55,56,57]. Data regarding vaccination recommendations are extrapolated from rheumatology guidelines, as most treatments addressed in this review overlap with drugs that are used for rheumatologic diseases.

## 3. Summary and Future Directions

This review (Table 2) highlights the increased risk of bacterial infections associated with the use of immunosuppressive agents that are widely employed for the treatment of a variety of interstitial lung diseases. Immune modulating drugs are the mainstay of treatment of interstitial lung diseases, especially for those that are associated with systemic rheumatologic conditions. Although the risk of infections with use of immune modulating agents is substantially high, and many can be associated with fatal infections, studies that have systematically addressed this issue are scant. Prospective longitudinal studies that allow us to investigate risk factors for infections in patients with ILD who are on immunosuppressive agents are urgently required to define the safety of immunosuppressive therapy, as well as the role of prophylaxis to inform future practical guidelines when these agents are used.

## Figures and Tables

**Table 1 pathogens-12-00464-t001:** Vaccinations and recommendations for patients receiving immunosuppression [54,55,56,57].

Vaccinations	Recommendations	Considerations
Non-live vaccine		
Pneumococcal vaccine	Recommended	Immunogenicity reduced by rituximab, abatacept and tofacitinib
Tdap	Recommended	
Meningococcal vaccine	Recommended	
Live vaccine	Not advised/recommended	Contraindicated with biologic therapy but may be considered with low dose immunosuppression

**Table 2 pathogens-12-00464-t002:** Summary of immunosuppressants and their mechanism of action leading to the increased susceptibility to bacterial infections.

Immunosuppression Used in ILD		Mechanism of Action of Immunosuppressants	Studies Showing Increased Risk of Infection
Steroid		Multiple anti-inflammatory and immunosuppressive effects and can affect any immune cell [13]	[17,19,20,21,22,23,29]
Steroid sparing agents	Azathioprine	Myelosuppression and inhibition of B and T cell proliferation [14,43]	[25,40,43]
	Mycophenolate Mofetil	Decrease T and B cell proliferation [14,44]	[41,44,45]
	Cyclophosphamide	Myelosuppression [14]	[14,46,47]
	Rituximab	B-cell depletion ± hypogammaglobulinemia [14]	[14,49]
	Abatacept	Inhibition of T-lymphocyte co-stimulation [50]	
	Tocilizumab	Anti-IL 6 monoclonal antibody [52]	[53]

## Data Availability

No new data were created or analyzed in this study. Data sharing is not applicable to this article.

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
