# Peer review of "Bacterial Infections Associated with Immunosuppressive Agents Commonly Used in Patients with Interstitial Lung Diseases"

_pathogens, 2023, doi:10.3390/pathogens12030464_

Round 1

Reviewer 1 Report

In this review, the authors have addressed an important topic- susceptibility of the patients diagnosed with interstitial lung disease (ILD) to bacterial infections when they are treated with immunosuppressive agents such as corticosteroids or other newly developed treatment regimens including treatment with steroid sparing agents like Rituximab. There is a plethora work on the effectiveness of steroids on ILD and its mechanism of action. Steroids or any other immunosuppressive agents acts as “necessary evil” for patients diagnosed with ILD. Although, treatments with immunosuppressive agents can cause decreased inflammation and suppression of tumor by inactivating the macrophage maturation, lowering the cell surface receptors of the immune cells, this also makes the patients more prone to bacterial infection. For example, Jick  et al have shown that treatment with corticosteroids have increased the chances of recurrent tuberculosis thereby worsening the patient condition.

The authors have also discussed about the effectiveness of newly developed treatment regimens including treatment with Rituximab that acts through B-cell depletion making the patients susceptible to bacterial infection or treatment with other steroid sparing agents including Cyclophosphamide that can cause myelosuppression or Abatacept that act by inhibiting the T-lymphocyte stimulation.

Overall, the review is well-written and should be informative to a vast majority of the scientist/clinicians who are involved with studying/treatment of patients with ILD. However, I’ve outlined my comments below that the authors should incorporate to enrich the review.

Major Comment

Although, the authors have addressed an important area of research and it a well-organized review, I felt that there was not emphasis given about the possible remedies/strategies that should be implemented to treat the patients with ILD. The authors should allocate a section discussing about the possible strategies the clinicians should adopt to minimize the risk of bacterial infections whole treating the patients with immunosuppressive compounds. Otherwise, this review feels like reading a bucket list of things that are already well known in the field.

Minor Comment

Line 172, there is a typo replace the word ndividuals with individuals.

Author Response

Major Comment

Although, the authors have addressed an important area of research and it a well-organized review, I felt that there was not emphasis given about the possible remedies/strategies that should be implemented to treat the patients with ILD. The authors should allocate a section discussing about the possible strategies the clinicians should adopt to minimize the risk of bacterial infections whole treating the patients with immunosuppressive compounds. Otherwise, this review feels like reading a bucket list of things that are already well known in the field. 

Response: Please see section on mitigation factors, and table 2

Minor Comment 

Line 172, there is a typo replace the word ndividuals with individuals. 

Response: Thank you. Corrected. 

Reviewer 2 Report

The review by Chaaban et al. aims to summarize the risk of bacterial infection in patients with ILD treated with immunosuppressive medications. The following is a synopsis of concerns raised based on the data presented in the manuscript.

Major:

1) The steroid section (2.1) focuses on the risk of the steroid itself. Little is discussed about steroid usage in the ILD patient population. 

2) In the sections on steroid-sparing agents, many of the cited articles did not specify how many patients have ILD (such as ref 47 is a study on lupus nephritis), particularly given that lung involvement is relatively uncommon in some CTDs (chronic polyarthritis). Also, the authors in the abstract mentioned that sarcoidosis is excluded from the review. However, ref 44 is on sarcoidosis. 

3) The table can be expanded to include the specifically cited studies showing an increased risk of infection. The authors can add a separate column after “mechanism of action of immunosuppressants.”

4) Many of the articles cited by the authors are reviews rather than research studies.

Minor:

1) A few places that the authors wrote “citation,” but no citations were listed

2) Several grammar/spelling mistakes, such as p4 line 172, “I” is missing from “individual.” 

Author Response

Major Comments

1) The steroid section (2.1) focuses on the risk of the steroid itself. Little is discussed about steroid usage in the ILD patient population. 

Response

We have revised this section.

Comment 2) In the sections on steroid-sparing agents, many of the cited articles did not specify how many patients have ILD (such as ref 47 is a study on lupus nephritis), particularly given that lung involvement is relatively uncommon in some CTDs (chronic polyarthritis). Also, the authors in the abstract mentioned that sarcoidosis is excluded from the review. However, ref 44 is on sarcoidosis. 

Response

It is important to emphasize that most of the literature used in this review is extrapolated from other diseases that utilize the same therapies that have been described in ILD. There is very scarce data and research on bacterial infection risk in ILD patients who are on immunosuppression. There also may be a confounding risk of the underlying disease risk for bacterial infection that cannot be elucidated without more trials.

The section on sarcoid was removed.

Comment 3) The table can be expanded to include the specifically cited studies showing an increased risk of infection. The authors can add a separate column after “mechanism of action of immunosuppressants.”

Response

Agreed. This has been added.

Comment 4) Many of the articles cited by the authors are reviews rather than research studies.

Response

We utilized this approach because there is a scarcity of data in the literature that specifically highlights bacterial infection in patients who are on immunosuppression.

Minor Comments

1) A few places that the authors wrote “citation,” but no citations were listed 

Response: corrected 

2) Several grammar/spelling mistakes, such as p4 line 172, “I” is missing from “individual.” 

Response: corrected 

Reviewer 3 Report

In their manuscript: Chhaban and Sadikot investigate an important issue of the increased risk of infections in patients treated for interstitial lung diseases.  There are some minor issues that need to be addressed

1.      A flow chart illustrating the decision process on accepting/rejecting articles (how many on what grounds) would be very helpful

2.      Not all citations are numbered (Verse 47,119,161,287)

3.      The paragraph from verse 97-105  would fit better into the part about steroids

Author Response

In their manuscript Chhaban and Sadikot investigate an important issue of the increased risk of infections in patients treated for interstitial lung diseases.  There are some minor issues that need to be addressed

Comment 1: A flow chart illustrating the decision process on accepting/rejecting articles (how many on what grounds) would be very helpful. 

Response

We now include this process in a Methods section. 

Comment 2

Not all citations are numbered (Verse 47,119,161,287)

Response

Corrected

Comment

The paragraph from verse 97-105 would fit better into the part about steroids.

Response

We agree. Thank you. This revision has been made. 

Round 2

Reviewer 1 Report

everything looks good

Author Response

Thank you.

Reviewer 2 Report

The authors addressed some of my concerns but again, the infection associated with the medications are not unique to ILD patients. I think the title should be changed to "bacterial infection associated with immunosuppressive agents commonly used in ILD"

Steroid are usually used for short-term treatment, at least in academic centers. The authors should provide more information on newer agents such as rituximab or tocillzumab. This is make the review more relevant and updated on current paradigm changes in CTD management.

Author Response

Comment: The authors addressed some of my concerns but again, the infection associated with the medications are not unique to ILD patients. I think the title should be changed to "bacterial infection associated with immunosuppressive agents commonly used in ILD"

Response: Thank you for the title suggestion, we adjusted the title as suggested to see fit with the content of the manuscript.

Steroid are usually used for short-term treatment, at least in academic centers. The authors should provide more information on newer agents such as rituximab or tocillzumab. This is make the review more relevant and updated on current paradigm changes in CTD management.

Thank you for your kind comment. The aim of the manuscript is to highlight the immunosuppressive medications used in ILD patient population and their associated bacterial infection risk and not to suggest treatment algorithms as there is center variability as was eluded in the comment on step up and step down of therapy.